# Research on the Impact of Water Conservancy Projects on Downstream Floodplain Wetlands—Taking Yimin River as an Example

**Chunming Hu and Xi Dong ***

State Key Laboratory of Urban and Regional Ecology, Research Center for Eco-Environmental Sciences, Chinese Academy of Sciences, Beijing 100085, China

* Correspondence: xidong@rcees.ac.cn

**Abstract:** Continued construction of reservoirs around the world promotes socio-economic development and severely affects the ecological and hydrological processes of rivers and floodplain wetlands. In this study, the Yimin River in Inner Mongolia, China, was taken as the research area. The water inundation guarantee rate (WIGR) was used as the model basis to characterize the inundation probability of the floodplain area. The comparative study of the remote sensing data of the 38 periods before the construction of the reservoir, and the 14 periods after the construction of the reservoir, shows that: due to the impact of the reservoir construction, the submerged area of the water body (WIGR greater than zero) decreased from 49.03 km$^2$ to 39.32 km$^2$, a total reduction of 9.71 km$^2$; the very low water inundation guarantee rate area (WIGR value of 0–20%) was the most affected, with a decrease of 12.14 km$^2$, while the area of other areas with a WIGR value greater than 20% increased by 3.43 km$^2$. In addition, the affected floodplain wetlands have significant spatial distribution characteristics: first, they are mainly distributed in the high-curvature river reach areas, and in this study 77.96% of the three high-curvature river reach areas accounted for 26.04% of the area, with area loss in very low WIGR areas; the second is that it is distributed far from the outside of the river channel, while the WIGR value in some areas near the river channel is increased. This study provides a technical reference for downstream wetland protection based on the WIGR model.

**Keywords:** floodplain wetland; water inundation guarantee rate; reservoir; Yimin River; remote sensing

## 1. Introduction

Low-lying areas with seasonal or intermittent water accumulation are defined as wetlands [1]. Wetlands are considered the most important and unique ecosystems in the world, with only 6% of the global wetland area, but the ecosystem services provided by wetlands account for 40% of the world's total [2,3]. After the 20th century, due to the rapid development of human society and the continuous progress of urbanization and industrialization, a large number of wetland ecosystems have been affected and destroyed, and global warming has also had a negative impact on the protection of wetlands. It is necessary to study and understand the change and impact mechanism of floodplain occurrence and development, and then formulate scientific and reasonable management policies to protect wetlands to prevent the loss of floodplain biodiversity and prevent habitat degradation [4,5].

With the increasing demand for hydropower resources in social and economic development, more and more water conservancy projects have been built and used in the world for hydropower generation and flood regulation [6]. The construction of water conservancy projects will change the hydrological conditions of the rivers, which in turn will have a significant impact on the wetlands around the rivers [7–9], such as loss of wetland landscape in floodplains, loss of biodiversity, and degradation of ecological functions that affects the horizontal and vertical ecological connections of rivers. It will cause a short circuit in the

hydrological cycle and thus cause the overall ecological environment of the region to be affected. Thus, destroying wetland habitats and wetland ecosystems, etc. [10–12]. Relevant scholars and research institutions have carried out much research on the impact of reservoir operations on wetland ecosystems, ecosystem protection, and restoration, and put forward many effective theoretical methods.

Advances in remote sensing and geographic information system technologies have promoted the development of wetland monitoring results, making it possible to conduct large-scale, long-term comprehensive monitoring of wetlands [13–15]. A large number of studies have focused on quantitative assessment of the impact of the construction of water conservancy projects on rivers, based on different methods such as field investigation, long-term series, and simulation analysis. Based on the existing mature remote sensing technology, modern scholars have developed a large number of models to quantify the relationship between floodplain areas and river flow, flood conditions, including integrated hydraulic and hydrological models [2,16,17], distributional explicit statistical models [18] and so on. For example, Zeng et al. [19] proposed a mix inexact-quadratic fuzzy water resources management model for floodplains (IQT-WMMF), which can provide an effective link between the system benefits and the relevant economic penalties caused by violating the predetermined water goals under the limited availability of data. The model makes reasonable use of water resources in the sustainable development of regional water resources. Zeng et al. [20] developed a simulated water environment management model based on Laplace Scenario Analysis (SWML) for planning regional sustainability in the composite wetland ecosystem. SWML can not only deal with spatial and temporal changes in hydrological elements in the basin, but also deal with various uncertainties in probability, probability distribution, and fuzzy sets. Zeng et al. [21] proposed an improved fuzzy approximate mixed random method (DFAS) for regional wetland ecosystem (RWE) management under uncertain conditions. The model can deal with traditional objective uncertainty, and the method can also reflect the decision-maker's risk attitude in the decision-making process. Zeng et al. [22] proposed a fuzzy random method based on the Hurwicz criterion (FSH) for land use planning of wetland ecosystems under uncertain conditions. This method can be used in actual land use management. We should advocate reasonable land use plans, improve ecological functions, and bring more intangible ecological benefits to human activities. Zuo et al. [23] selected patch density, maximum patch index and contagion index as indicators to measure the protection effect of Zoige Wetland National Nature Reserves (WNNRs). Lu et al. [24] applied Landsat Thematic Mapper image data and maximum entropy model to analyze the protective effects of wetland reserves in Northeast China. In 2021 Dong et al. [25] based on long-term remote sensing data and hydrological data, proposed a new assessment method for describing the floodplain inundation status of river floodplain wetlands, the Water Inundation Guaranteed Rate (WIGR), which makes the quantification of wetland inundation status more scientific and accurate. Zeng et al. [26] designed a wetland reconfiguration (WR) plan based on water indicators to deal with various ecological crises and flood risks in the Yongding River floodplain in China. Among them, farmland can be considered to be converted into wetlands (CFW) and water diversion (WD) to balance the trade-off between artificial land use and wetland protection. The above methods are as follows: research on monitoring floodplain wetlands provides ideas. However, from the perspective of regional scope, the horizontal and vertical impacts of water conservancy project construction on floodplain wetlands and the key roles of rainfall and runoff in the process of floodplain wetland inundation state changes are rarely studied.

China is one of the countries with rich ecosystems in the world, with abundant ecological resources such as wetlands, forests, and grasslands [27]. This study takes the Yimin River in the Inner Mongolia Autonomous Region of China as the research object, combines remote sensing data with surface hydrological observation data, and deeply explores the horizontal and vertical impacts of the construction of Honghuaerji Reservoir on the floodplain wetlands on both sides of the river. This study hopes to achieve the

following main goals: (1) to explore the relationship between rainfall, hydrological regime, and inundation status; (2) to quantitatively identify the degree of change in the submerged status of rivers and wetlands before and after the construction of the reservoir; (3) to accurately reveal the reservoir status Construction of significant areas of impact on wetlands. We hope that this study can provide theoretical support for floodplain wetland managers.

## 2. Study Area and Data

### 2.1. Study Area

The study area is located in the hinterland of Hulunbuir grassland, Inner Mongolia Autonomous Region, China, as shown in Figure 1. The Yimin River originates from the Daxing'an Mountains, with a river length of 359.4 km and a drainage area of 22,640 km$^2$. The average annual runoff is 1.275 billion m3. The floodplain wetlands on both sides of the river are well developed. In 2010, the Honghuaerji Reservoir on the mainstream of the Yimin River was built to store water.

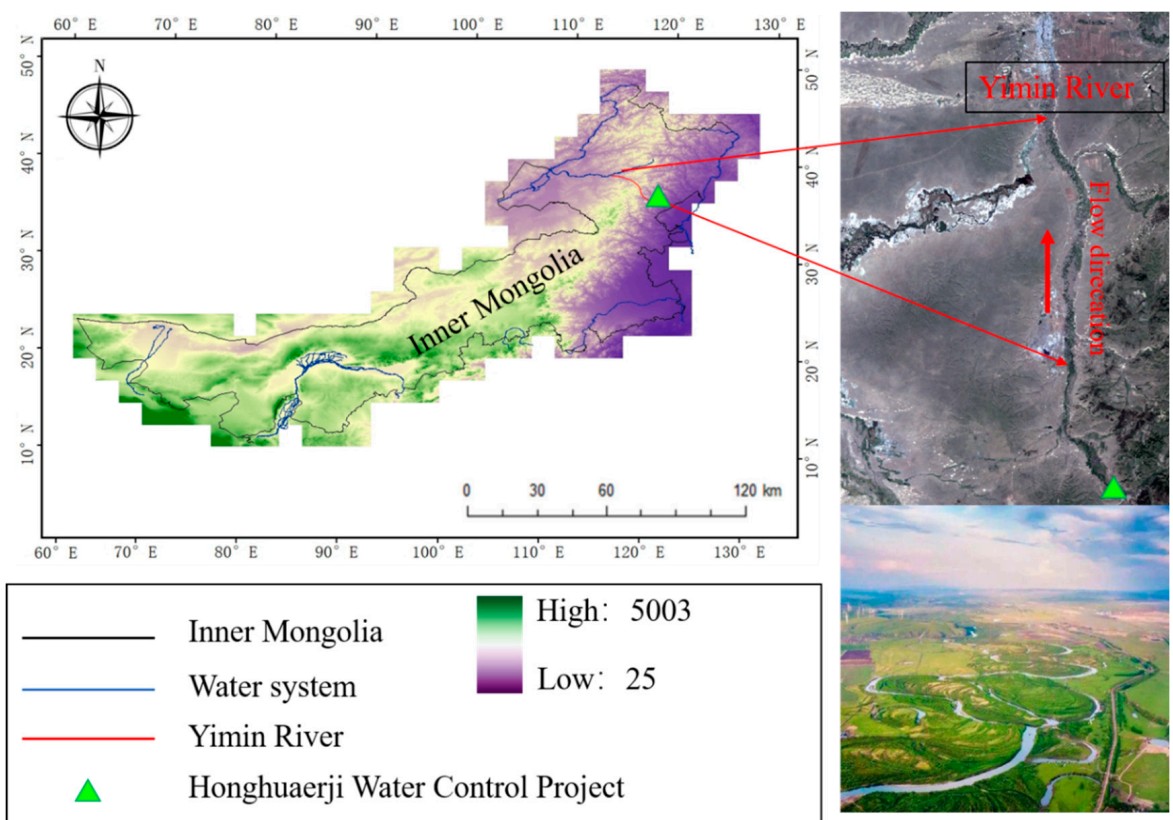

**Figure 1.** Study area location.

The Yimin River is a first-class tributary of the Hailar River. It flows from south to north, runs through the Ewenki Autonomous Banner, passes through the urban area of Hailar District, and joins the Hailar River in the north of Hailar District. The Yimin River is located in the transition zone from the mountains to the grasslands in the southeastern part of the Hulunbuir Grassland, and its tributaries are mainly distributed on the right bank.

In the upper reaches of Yimin River, the area above Honghuaerji, the river is 20–50 m wide, the forest is luxuriant, and Pinus sylvestris is the dominant tree species. Below Honghuaerji, the river enters the hills and grasslands. The width of the river is 50 to 80 m. The floodplain wetlands on both sides of the downstream river are well developed, which means there are many kinds of vegetation and many biological species in the wetland, and it can be frequently flooded by runoff or rainwater. The floodplain wetlands and swamps are scattered around the riverbank. It is easy to spread to the two banks to form wetlands, and the tributaries on both banks are densely covered. The study area has a special position

in regional ecological environment protection and has various ecological service functions such as ecological bank protection, groundwater recharge, and biodiversity support.

## 2.2. Data Source and Processing

The daily flow data of the Yimin River from 1986–2019 was obtained from the Hydrological Survey Administration. The monthly rainfall data of the study area from 1986–2019 were collected from the China Meteorological Data Sharing Website (http://www.nmic.cn/. Accessed on 6 May 2021). The Landsat5 TM/8 OLI-TIRS 1T-level images that were used in this study were acquired from the National Aeronautics and Space Administration (NASA) of the United States (Table 1). The images covered the entire study area, and their serial number was 123026. All available images during the plant-growth period (May–October) from 1986–2019 were acquired, and all remote sensing data with cloud amounts smaller than 2% were screened in this study. A total of 52 periods of available Landsat5 ETM/OLI data from May–October from 1986–2019 was collected, including 38 periods (multiple images in a period) before dam construction (1986–2009) and 14 periods after dam construction (2010–2019).

**Table 1.** Landsat5 TM/8 OLI data situation.

| Satellite | Band | Nominal Spectral Location | Satellite | Band | Nominal Spectral Location |
|---|---|---|---|---|---|
| | 1 | Blue | | 1 | Costal/Aerosol |
| | 2 | Green | | 2 | Blue |
| | 3 | Red | | 3 | Green |
| | 4 | Near IR | | 4 | Red |
| Landsat5 TM | 5 | SWIR-1 | Landsat8 OLI | 5 | Near IR |
| | 6 | LWIR | | 6 | SWIR-1 |
| | 7 | SWIR | | 7 | SWIR-2 |
| | | | | 8 | Panchromatic |
| | | | | 9 | Costal/Aerosol |

The irradiance was calculated using radiometric calibration in ENVI 5.3 and performed atmospheric correction of the data. The radiation document was input into the FLAASH atmospheric calibration module (ACM) in ENVI5.3. The ACM inputs automatically matched the calibration parameters from the radiation images, while other parameters (such as the date and time of flight and ground elevation) had to be input manually. The atmospheric model was set as the temperate zone. The Aerosol model was set to 'Rural', and the Aerosol retrieval was automatically set to the '2-waveband Kaufman-Tenley (K-T) method'. The input parameters varied according to the scene. All reflectivity products had values between 0 and 1. Further, accurate geometric calibration was performed on all reflectivity products. All of the 10 selected calibration points were distributed uniformly in the experimental region of the images, and the error was smaller than 1. Next, the water body index was calculated using the waveband computing tool.

## 3. Methodology

### 3.1. Technical Process

The research roadmap is shown in Figure 2. By extracting the water body based on the MNDWI water body index from the 52-phase available remote sensing data in the study area, the water body inundation area was determined. Based on Matlab and combined with the flow frequency statistics, the water body inundation frequency of all available images was calculated pixel by pixel, and further discussed for the influence of rainfall and runoff on the submerged state and the horizontal and vertical influence of the construction of the water conservancy project on the floodplain wetland.

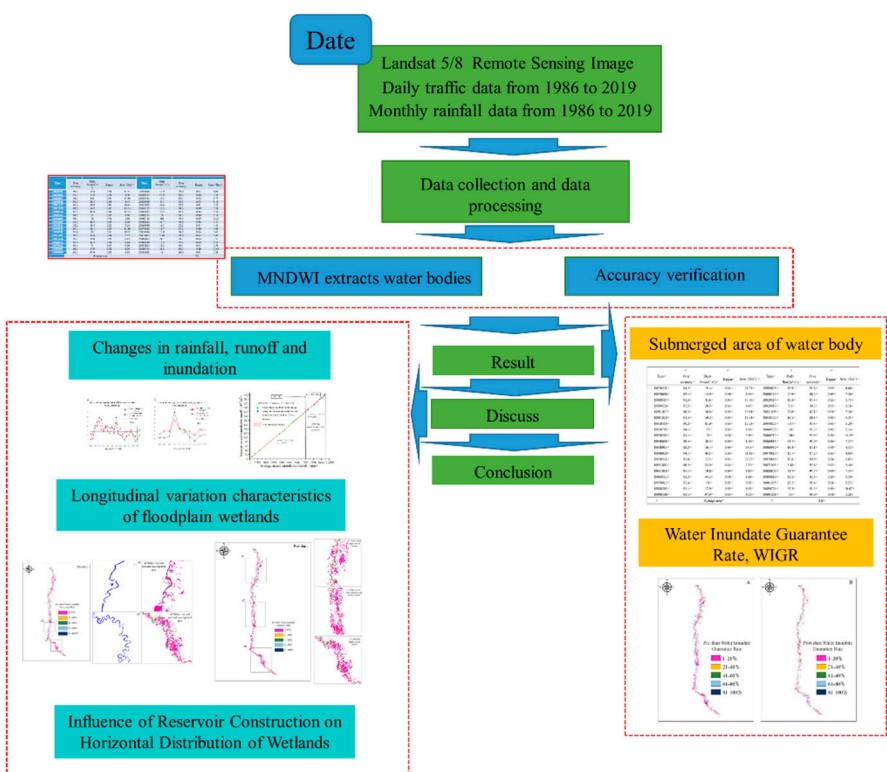

**Figure 2.** Flow chart.

### 3.2. Water-Body Extraction Method

Various remote sensing indexes could be applied to extract water bodies and wetlands mapped from satellite images, including NDWI [15], MNDWI [28], AWEI [29], and the improved normalized difference water index (RmNDWI) [13]. In this study, the MNDWI was used to draw water-inundation diagrams and analyze the wetland area of water inundation. MNDWI is Modified NDWI, which means to improve the normalized difference water index [28]. To verify water-inundation diagrams, the accuracy of wetland regions was proven using a visual interpretation of sampling points (50 points) and relatively high-resolution Google Earth images (50 points). Moreover, the water-body extraction method based on the computation of the Kappa coefficient and MNDWI could help ascertain the wetland state at both banks of the Yimin River.

$$MNDWI = (Green - MIR)/(Green + MIR) \tag{1}$$

where Green, NIR, and MIR refer to the green, near-infrared, and middle-infrared bands of the Landsat images, respectively. If the MNDWI value is higher than 0, that area contains a water body.

A quantitative assessment regarding the accuracy of the study area was carried out. The accuracy of water-body extraction was assessed based on Kappa coefficients (Foody, 2002). The accuracy is higher if the Kappa coefficient is closer to 1, which can be calculated as follows:

$$Kappa = \frac{P \sum_{i=1}^{n} P_{ii} - \sum_{i=1}^{n} (P_{i+} \times P_{+i})}{P^2 - \sum_{i=1}^{n} (P_{i+} \times P_{+i})} \tag{2}$$

where $P$ is the total number of pixels in the reference data, $P_{ii}$ is the total number of accurate pixels in different classes, $P_{i+}$ is the total number of pixels of classes in the classification data, $P_{+i}$ is the total number of pixels of classes in the reference data, and n is the total number of classes.

### 3.3. Water Inundation Guarantee Rate

Based on the concept of water inundation guarantee rate (WIGR), the WIF index (inundation frequency, that is, the number of times wetlands are flooded by river water per unit time) is used to study the health and stability of seasonally flooded wetlands [8,30]. Limited by spectral index data sources, the amount of image data used for research is often limited, which makes it difficult to scientifically characterize the actual long-term situation of wetlands. The traditional method for calculating floodplain wetland inundation state does not consider the impact of the hydrological regime, and the accuracy of the inundation state coverage area obtained under different frequencies is limited. The floodplain wetland inundation state is closely related to the hydrological regime of the river. Dong et al. [25] propose a WIGR model based on this feature of the wetland, and combining WIF [7] and WIGR can improve the accuracy and scientificity of determining the flooded area. The Matlab platform is used to realize the calculation of Equation (4). The ecological significance of WIGR is to accurately describe the frequency of floodplain wetlands inundating both banks of the river; that is, the number of times wetlands are inundated. The mathematical expression for WIGR is

$$\text{WIGR}(x_i, y_i) = \frac{\sum_{j=1}^{z} R_j \times P_j}{\sum_{j=1}^{z} P_j} \tag{3}$$

$$R_j = \begin{cases} = 1 \\ = 0 \end{cases}$$

The parameters selected by the model developer include the total number of remote sensing images used for remote sensing data and hydrological data, the number of remote sensing images with water pixels, the assurance rate of hydrological data and the total amount of flow data. Specific parameters are described as follows: where $(x_i, y_i)$ is a pixel in the image, $R_j$ is the water inundation of this pixel in the image at period $j$ (the value is 1 when the wetland is submerged by the river; the value is 0 when the wetland is not submerged by the river), $P_j$ is the guarantee of the flow rate on the day of the image at period $j$ in the historical daily average flow statistics, and Z is the total number of periods of all remote sensing images in this study.

Based on the WIGR value, all pixels were divided into five types according to their water inundation level: extremely low, low, medium, high, and extremely high.

### 3.4. Double Cumulative Curve Method

In order to explore the correlation between rainfall, runoff and wetland inundation state, we introduced the double cumulative curve method. The consistency analysis of the hydrological sequence can be characterized by the offset characteristics of the double cumulative curve. The change in the shape characteristic of the cumulative curve represents the change in the hydrological sequence under different influence conditions. The basic principle is to use the statistical data between the hydrological elements to establish Influence relationship between human activities and hydrological processes. The linear relationship between various hydrological elements in the base period can be expressed by the following formula [31]:

$$\sum Q = k \sum P + b \tag{4}$$

In the formula, $\sum Q$—cumulative runoff; $\sum P$—cumulative rainfall; $k$, $b$—calculation parameters.

## 4. Results

### 4.1. Water-Inundation Area

The water-inundation area in the study area was extracted using the MNDWI [28]. The Kappa coefficient indicates that the overall accuracy of water-body extraction is relatively high, i.e., higher than 90%.

A total of 38 periods of image data before dam construction were analyzed (Table 2). The average water inundation area was 8.6 km$^2$. The maximum was 29.22 km$^2$ on 11 May 1995, and the minimum was 3.18 km$^2$ on 19 September 2002.

**Table 2.** Verification of the accuracy of the MNDWI, Kappa coefficient, and water-inundation area in 38 periods before dam construction [25].

| Time | Accuracy | Daily Flow (m$^3 \cdot$s$^{-1}$) | Kappa | Area (km$^2$) | Time | Daily Flow (m$^3 \cdot$s$^{-1}$) | Accuracy | Kappa | Area (km$^2$) |
|---|---|---|---|---|---|---|---|---|---|
| 19870521 | 94.2 | 19.4 | 0.94 | 11.23 | 19990826 | 13.9 | 91.8 | 0.93 | 6.66 |
| 19870606 | 92.1 | 13.9 | 0.91 | 6.70 | 20000524 | 57.9 | 88.2 | 0.86 | 7.36 |
| 19880507 | 91.3 | 103 | 0.90 | 17.50 | 20020514 | 19.3 | 85.4 | 0.92 | 5.75 |
| 19890526 | 95.5 | 28.5 | 0.96 | 6.45 | 20020919 | 9.3 | 90.3 | 0.93 | 3.18 |
| 19891017 | 96.3 | 30.8 | 0.96 | 20.64 | 20031008 | 29.6 | 82.1 | 0.91 | 7.26 |
| 19901020 | 91.5 | 59.2 | 0.90 | 13.53 | 20040722 | 16.5 | 88.3 | 0.90 | 5.51 |
| 19910516 | 90.2 | 81.9 | 0.89 | 12.53 | 20040823 | 7.67 | 95.4 | 0.96 | 5.29 |
| 19920705 | 94.1 | 55 | 0.96 | 8.98 | 20060525 | 16 | 91.2 | 0.90 | 5.14 |
| 19930505 | 91.1 | 28 | 0.91 | 5.60 | 20060712 | 108 | 93.9 | 0.89 | 14.29 |
| 19940606 | 89.5 | 48.3 | 0.88 | 8.26 | 20060813 | 44.7 | 91.0 | 0.88 | 7.37 |
| 19940913 | 88.2 | 56.4 | 0.86 | 9.05 | 20060930 | 16.3 | 85.1 | 0.84 | 4.32 |
| 19940929 | 94.1 | 46.1 | 0.89 | 11.36 | 20070613 | 16.7 | 87.2 | 0.86 | 4.60 |
| 19950511 | 93.6 | 207 | 0.60 | 29.22 | 20070816 | 10.8 | 89.9 | 0.84 | 5.65 |
| 19951002 | 96.5 | 24.9 | 0.94 | 7.52 | 20071003 | 5.18 | 91.6 | 0.92 | 5.40 |
| 19951018 | 95.1 | 19.8 | 0.96 | 6.12 | 20080802 | 21.7 | 91.7 | 0.94 | 7.53 |
| 19960513 | 93.2 | 44.2 | 0.91 | 9.14 | 20080919 | 13.2 | 92.4 | 0.89 | 5.50 |
| 19970612 | 92.4 | 91 | 0.89 | 9.00 | 20081005 | 13.2 | 95.4 | 0.94 | 5.53 |
| 19980503 | 91.1 | 17.9 | 0.88 | 6.00 | 20090704 | 53.3 | 91.5 | 0.89 | 10.67 |
| 19990506 | 92.1 | 97.9 | 0.92 | 6.02 | 20091008 | 13 | 93.0 | 0.91 | 5.28 |
| Average area | | | | | | | | | 8.6 |

A total of 14 periods of image data after dam construction were analyzed (Table 3). The average water-inundation area was 11.24 km$^2$. The maximum was 25.19 km$^2$ on 2 September 2019, and the minimum was 4.49 km$^2$ on 1 October 2018.

**Table 3.** Verification of the accuracy of the MNDWI, Kappa coefficient and water-inundation area in 14 periods after dam construction [25].

| Phases | Time | Daily Flow (m$^3 \cdot$s$^{-1}$) | Over Accuracy | Kappa | Area (km$^2$) |
|---|---|---|---|---|---|
| | 2010504 | 97.5 | 89.7 | 0.88 | 11.23 |
| | 2010723 | 7.98 | 88.6 | 0.86 | 7.36 |
| | 2010909 | 10.4 | 84.6 | 0.83 | 4.93 |
| | 2011811 | 81.4 | 94.1 | 0.89 | 9.35 |
| | 2011912 | 13.8 | 93.6 | 0.60 | 5.11 |
| | 20131003 | 56.6 | 96.5 | 0.94 | 16.35 |
| | 2014531 | 87.9 | 95.1 | 0.96 | 21.62 |
| Post Dam | 2015708 | 66.9 | 93.2 | 0.91 | 16.17 |
| | 20151025 | 23.4 | 92.4 | 0.89 | 9.38 |
| | 2016504 | 69.7 | 91.1 | 0.88 | 13.45 |
| | 2017624 | 18.2 | 92.1 | 0.92 | 8.02 |
| | 20171030 | 2.23 | 91.8 | 0.93 | 4.68 |
| | 20181001 | 2.31 | 88.2 | 0.86 | 4.49 |
| | 2019902 | 195 | 94.1 | 0.89 | 25.19 |
| Average area | | | | | 11.24 |

### 4.2. Water Inundation Guarantee Rate

The development of floodplain wetlands is closely related to the hydrological conditions including river flow [18]. The WIGR of the study area is shown in Figure 3. The area statistics in different WIGR regions are listed in Table 4.

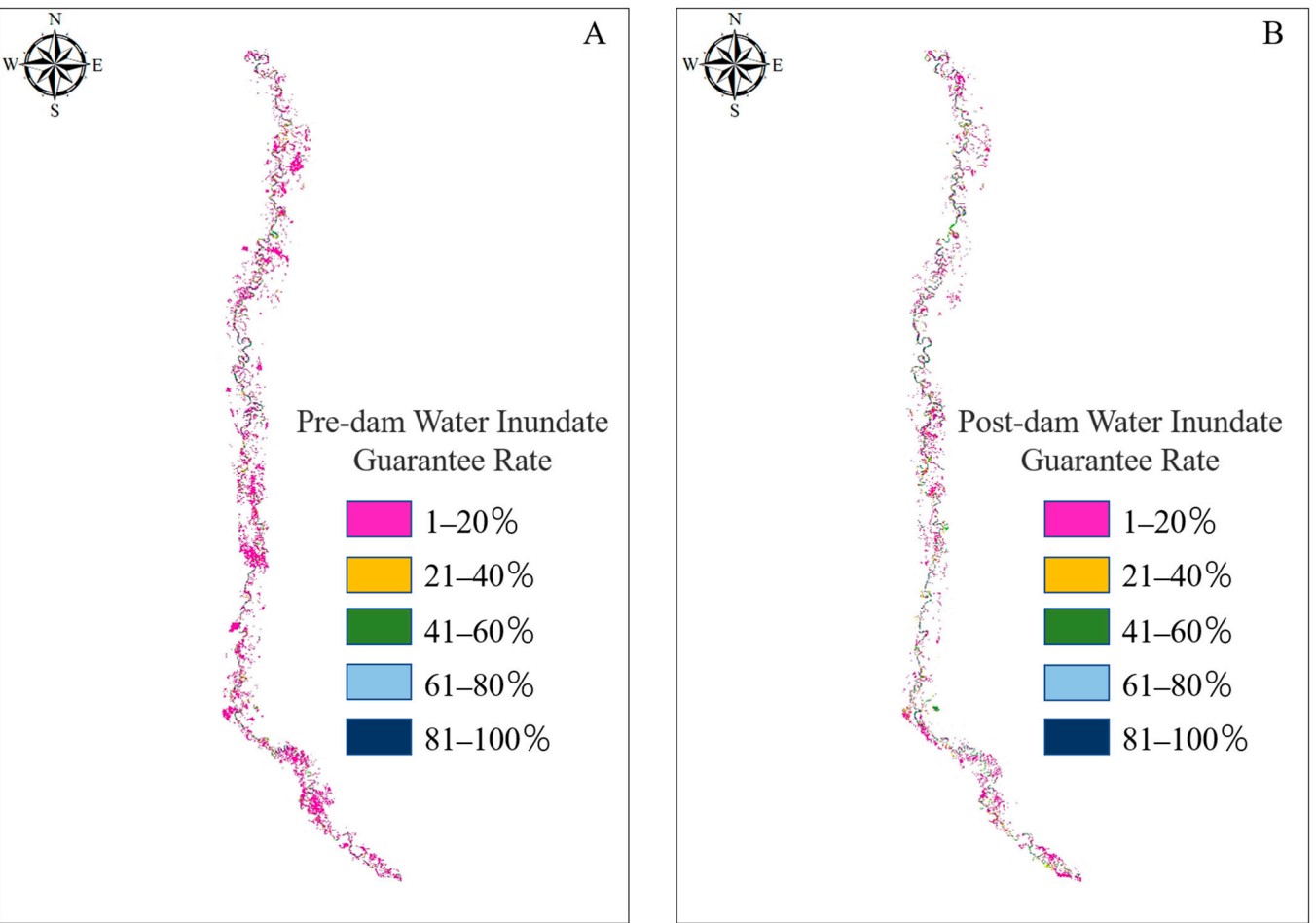

**Figure 3.** Statistical diagram of WIGR distribution in the study area ((**A**) pre-dam construction; (**B**) post-dam construction).

**Table 4.** Area statistics for different WIGR values before and after dam construction [25].

| WIGR (%) | Pre-Dam | | Post-Dam | | Area Change in km² |
|---|---|---|---|---|---|
| | Area in km² | % of Area | Area in km² | % of Area | |
| 0–20 | 38.71 | 78.94 | 26.57 | 67.58 | −12.14 |
| 21–40 | 3.64 | 7.42 | 4.39 | 11.18 | 0.75 |
| 41–60 | 2.24 | 4.57 | 3.11 | 7.91 | 0.87 |
| 61–80 | 1.82 | 3.71 | 2.26 | 5.76 | 0.44 |
| 81–100 | 2.63 | 5.36 | 2.98 | 7.58 | 0.35 |
| Total | 49.03 | 100 | 39.32 | 100 | −9.71 |

The WIGR distribution and variation in different regions remained fairly consistent before and after dam construction, other than some differences in numerical values. The areas with extremely low WIGR values were influenced the most by dam construction. The area with extremely low WIGR after dam construction decreased by 12.14 km², while areas with low, medium, high, and extremely high WIGR increased by 0.75 km², 0.87 km², 0.44 km², and 0.35 km², respectively.

## 5. Discussion

### 5.1. Evaluation System

Based on remote sensing data, hydrology, and rainfall data, our research analyzed the relationship between rainfall and runoff, as well as the spatial distribution of floodplain wetlands caused by the construction of water conservancy projects. Therefore, we built the following evaluation system, as shown in Figure 4. We drew the WIGR map of Yimin River, and then we analyzed the relationship between rainfall and hydrology in the study area in the past 30 years and the construction of water conservancy projects, which led to changes in runoff. It has an impact on the horizontal and vertical inundation state of the floodplain wetland in the study area.

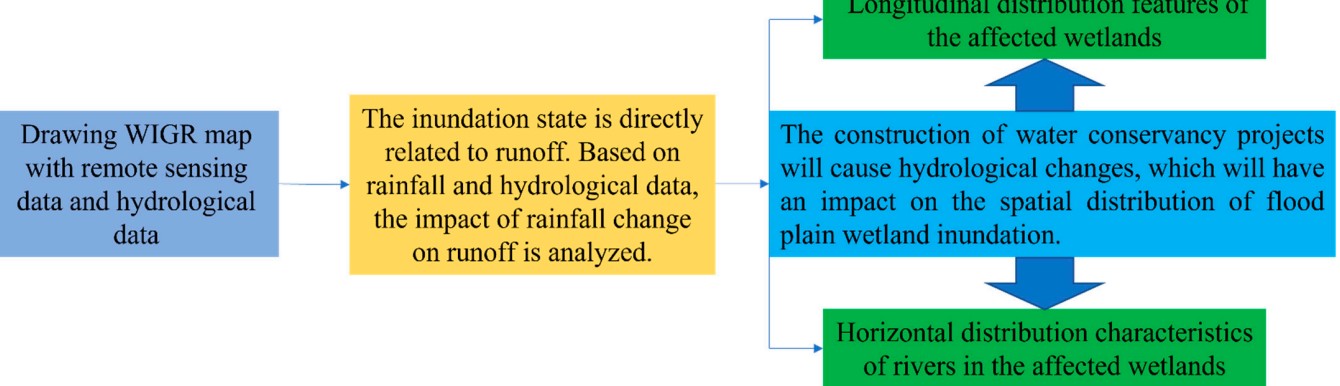

**Figure 4.** Evaluation system process.

### 5.2. Changes in Rainfall, Runoff and Inundation

As shown in Figure 5 in this study, the annual average runoff and rainfall data before and after the construction of the reservoir were compiled. The changing trend of the two is the same, and there is a significant correlation between the hydrological regime and the rainfall change. In the absence of external interference, rainfall changes directly act on river runoff, and these rainfall changes alter the hydrological cycle of the river system by changing the magnitude, duration, frequency, timing, predictability, etc., of hydrological process events.

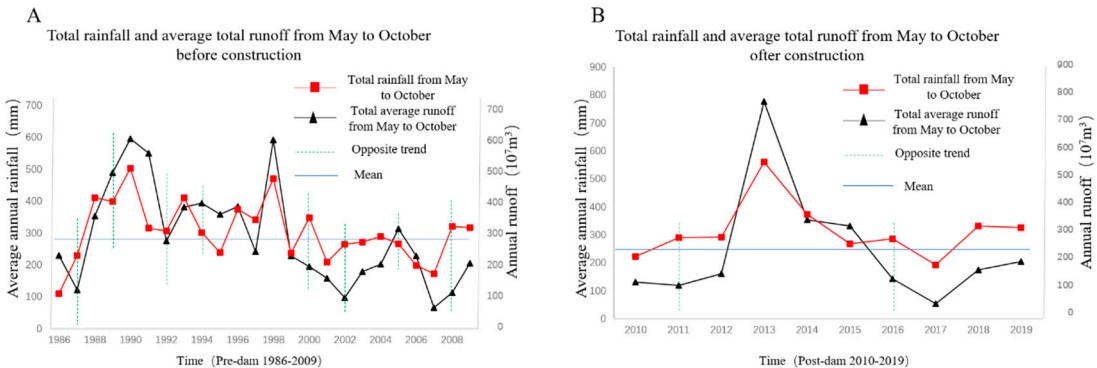

**Figure 5.** Trends of rainfall and runoff before and after the construction of the water conservancy project.

The runoff of the Yimin River comes from precipitation and snowmelt supply [31]. According to the double cumulative curve of precipitation and runoff at Yimin Ranch Station from 1986 to 2019 (see Figure 6), there is a good linear relationship between precipitation and runoff. Based on the construction time of the Honghuaerji Water Control Project in 2009, the research period is divided into two periods.

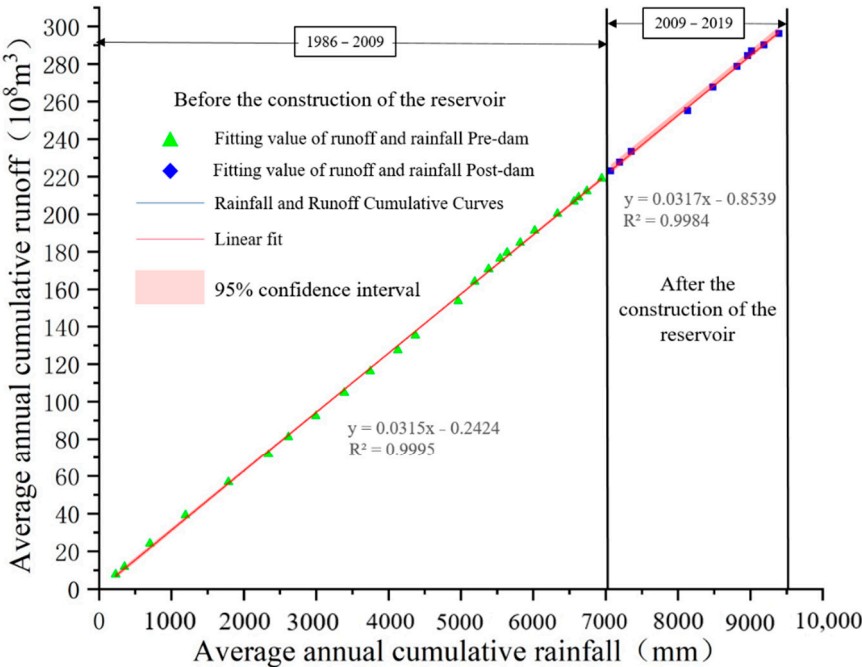

**Figure 6.** The double cumulative curve of precipitation and runoff from 1986 to 2019 at the Yimin Ranch Hydrological Station.

From 1986 to 2009, the correlation between runoff and precipitation was excellent and the coefficient of determination $R^2$ of linear fitting was 0.9995, indicating that rainfall directly affected runoff and other disturbances were less.

From 2009 to 2019, the construction of the Honghuaerji Water Control Project was completed and put into use. The runoff process was affected to a certain extent, but the response relationship between precipitation and runoff was still good.

Affected by the construction of the water conservancy project, the runoff in the river channel was relatively reduced, the determination coefficient $R^2$ of the precipitation-runoff fitting line was also slightly reduced to 0.9984, and the 95% prediction interval of the cumulative runoff estimated based on the cumulative precipitation was also significantly wider than the previous stage, indicating that the predictability of runoff becomes worse.

Rainfall and snowmelt can regulate river runoff and affect the river's water supply to floodplain wetlands [4,32,33]. The mutual transformation of water, salinity, and nutrients in floodplain wetlands between surface water and groundwater are closely related [30]. The construction of the water conservancy project will lead to insufficient water supply in the floodplain, which will reduce the horizontal connection between the floodplain and the river, thereby reducing the number of floods overflowing the river channel, reducing the frequency of floodplain wetland inundation, and affecting the development of floodplain wetlands, thus affecting the ecological environment. System service function produces an inhibitory effect.

*5.3. Longitudinal Distribution Features of the Affected Wetlands*

The ground response to dams varies with location, environment, substrate, inundation conditions, and sediment loading [33], and the magnitude of the impact is usually shaped by the combined effects of various driving factors, which also lead to difficulty assessing the impact of dam construction on floodplain wetlands [12]; Existing studies have shown that dam construction will reduce the frequency of flooding along the river, and at the same time the degree of flooding will be reduced, thus affecting the floodplain wetland inundation [8]. This study focuses on evaluating river bends, where high-curved river sections and straight river sections before and after the construction of the Yimin River reservoir are randomly selected. Through our research we found that the development of

floodplain wetlands in high curvature main rivers is due to the floodplain wetlands on both sides of straight rivers—see Figures 7 and 8. a1 and a2, respectively, representing the development of floodplain wetlands in the main channel and straight section of the Yimin River in the high bend section before the construction of the water conservancy project, and b1 and b2, respectively, representing the development of floodplain wetlands in the main channel and straight section of the Yimin River in the high bend section after the construction of the water conservancy project. The inundation state of the highly curved river segment and the straight river segment are different, and the development state of the floodplain wetland is different. The shape of the high curvature river segment determines the flow conditions in the river channel and the development degree of the floodplain wetland [34]. In our study we only qualitatively describe the degree of curvature of the river and the floodplain research, and quantitative research will be the focus of our next research [35].

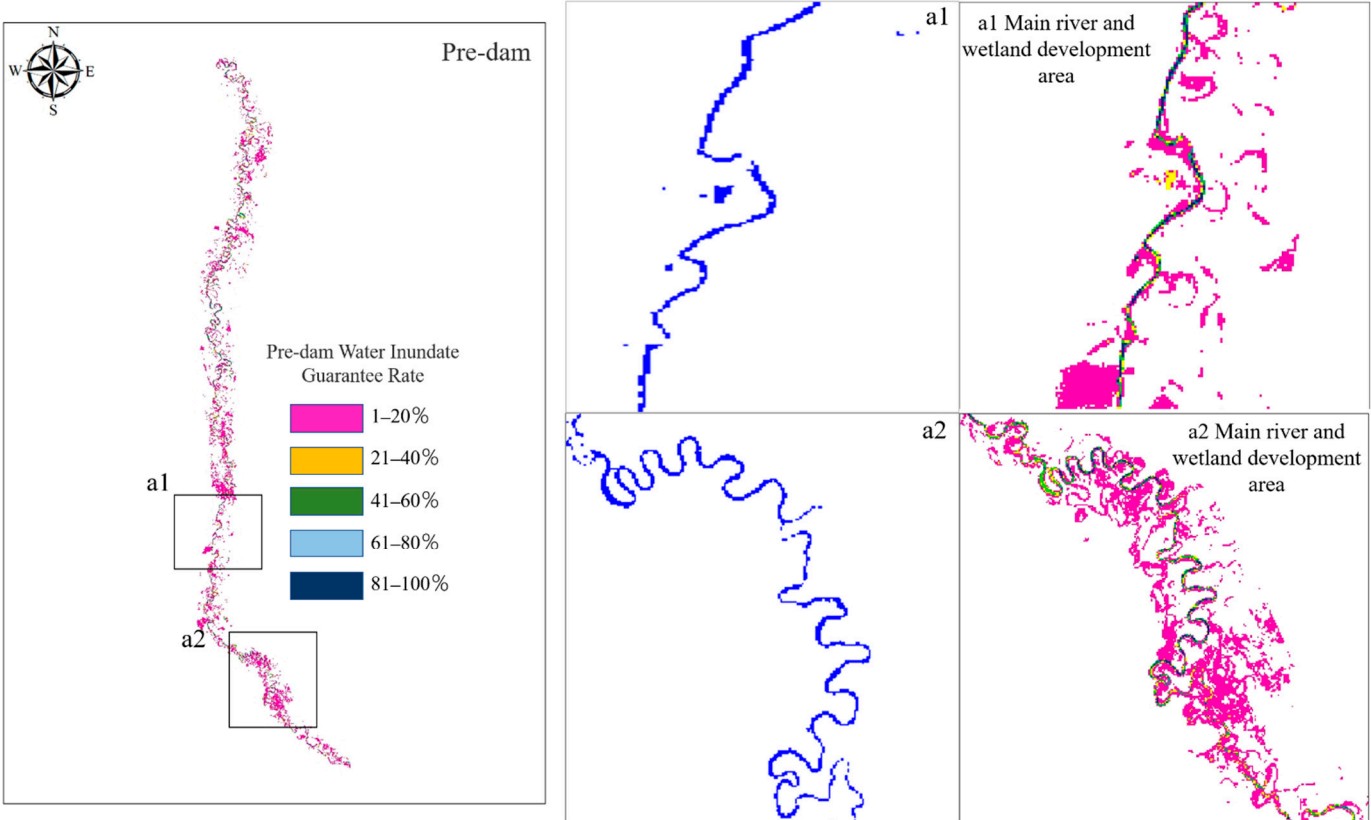

**Figure 7.** Pre-dam differences the submerged state between the highly curved river segment and the straight river segment.

In this study, three high-curvature river sections in the region were selected to explore the influence of the degree of river curvature on the wetland inundation state. Figures 9 and 10 indicates the development of floodplain wetlands on both sides of the main river channel in highly curved areas before and after the construction of the water conservancy project, showing the same significant characteristics. Some studies have demonstrated that the bending degree of the watercourse affects the WIGR of surrounding wetlands to some extent [36]. The results obtained before dam construction are listed in Table 5. The WIGR area in the three reaches was 20.70 km$^2$, which accounted for 42.22% of the total WIGR area within the study area. However, the areas of the three reaches only accounted for 26.04% of the total study area. After dam construction the WIGR area in the three reaches decreased by 9.47 km$^2$, which accounted for 77.96% of 12.14 km$^2$. It shows that the three regions have a high degree of river curvature, a high frequency of wetlands

overflowing and inundation by river runoff, and floodplain wetland vegetation due to frequent inundation by runoff where wetlands develop best, but these areas are also most significantly affected by reservoir construction.

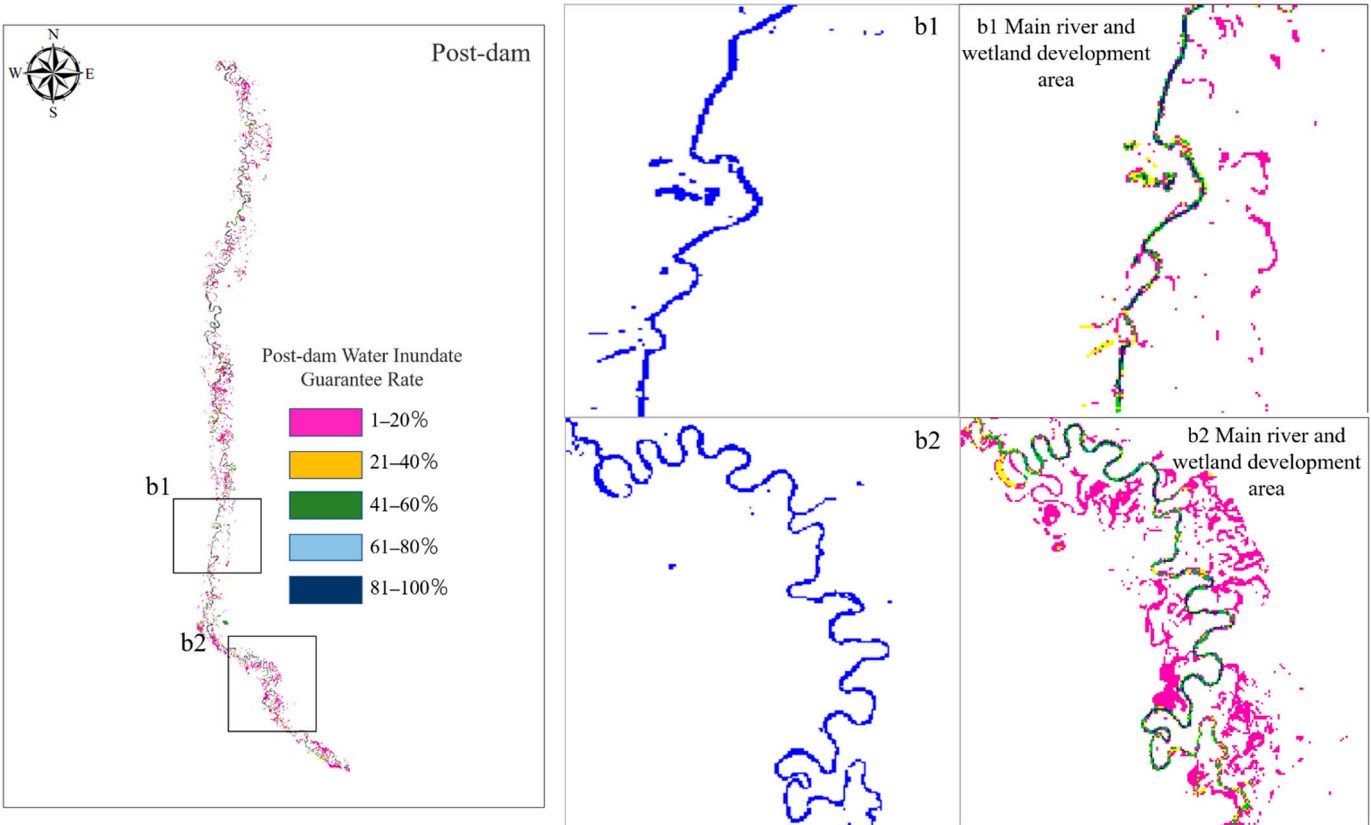

**Figure 8.** Post-dam differences the submerged state between the highly curved river segment and the straight river segment.

**Table 5.** Typical regional variation area difference in WIGR.

| Spatial Location | Different Frequency Range | Pre-Dam Area in km² | Post-Dam Area in km² | Area Change in km² |
|---|---|---|---|---|
| High curvature channel | 0–20 | 17.22 | 7.76 | −9.46 |
| | 20–40 | 1.41 | 1.59 | 0.18 |
| | 40–60 | 0.81 | 1.08 | 0.27 |
| | 60–80 | 0.63 | 0.77 | 0.14 |
| | 80–100 | 0.62 | 0.85 | 0.23 |
| | Total | 20.7 | 12.05 | −8.65 |

In general, three regions had high curvature, high WIGR, and the best development characteristics for floodplain wetlands. However, these regions were influenced the most by dam construction. In regions with high curvature, the water exchange between the floodplain and main channel produced a tributary with horizontal flow speed. Under different water-inundation conditions, flows along the vertical and horizontal directions were reoriented [37]. Water-flow dynamics exist along different directions in rivers with high curvature. For rivers in regions with high WIGR, the floodplain wetlands have good connectivity with water bodies in the main rivers, which is influenced by frequent water inundation [38]. This can ensure the supply of essential water for vegetation growth in the region, allowing floodplain wetlands to develop fully. This conforms to the research results of Karim et al. [39].

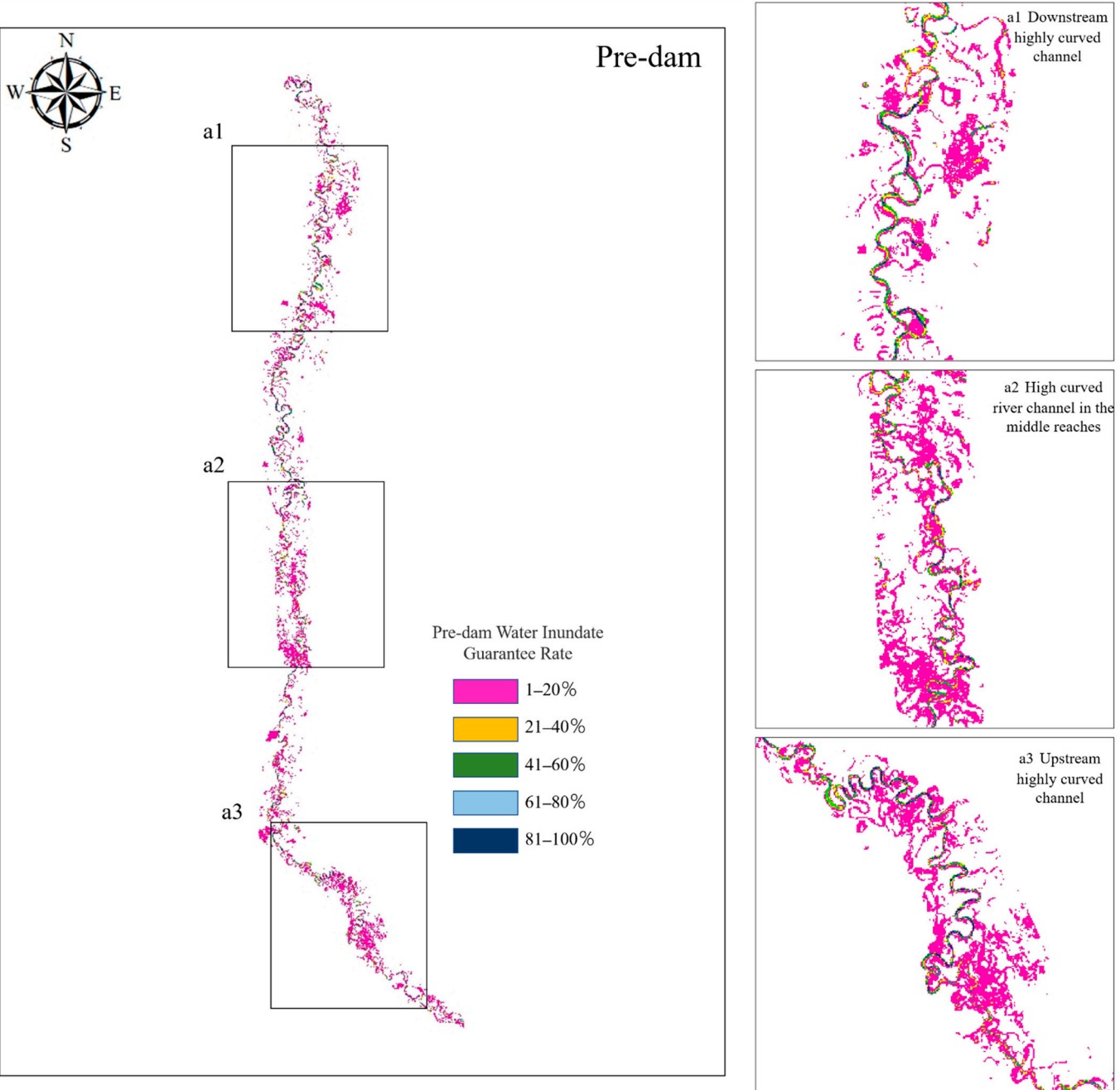

**Figure 9.** Pre-dam area distribution and typical change area under different inundation guarantee rates.

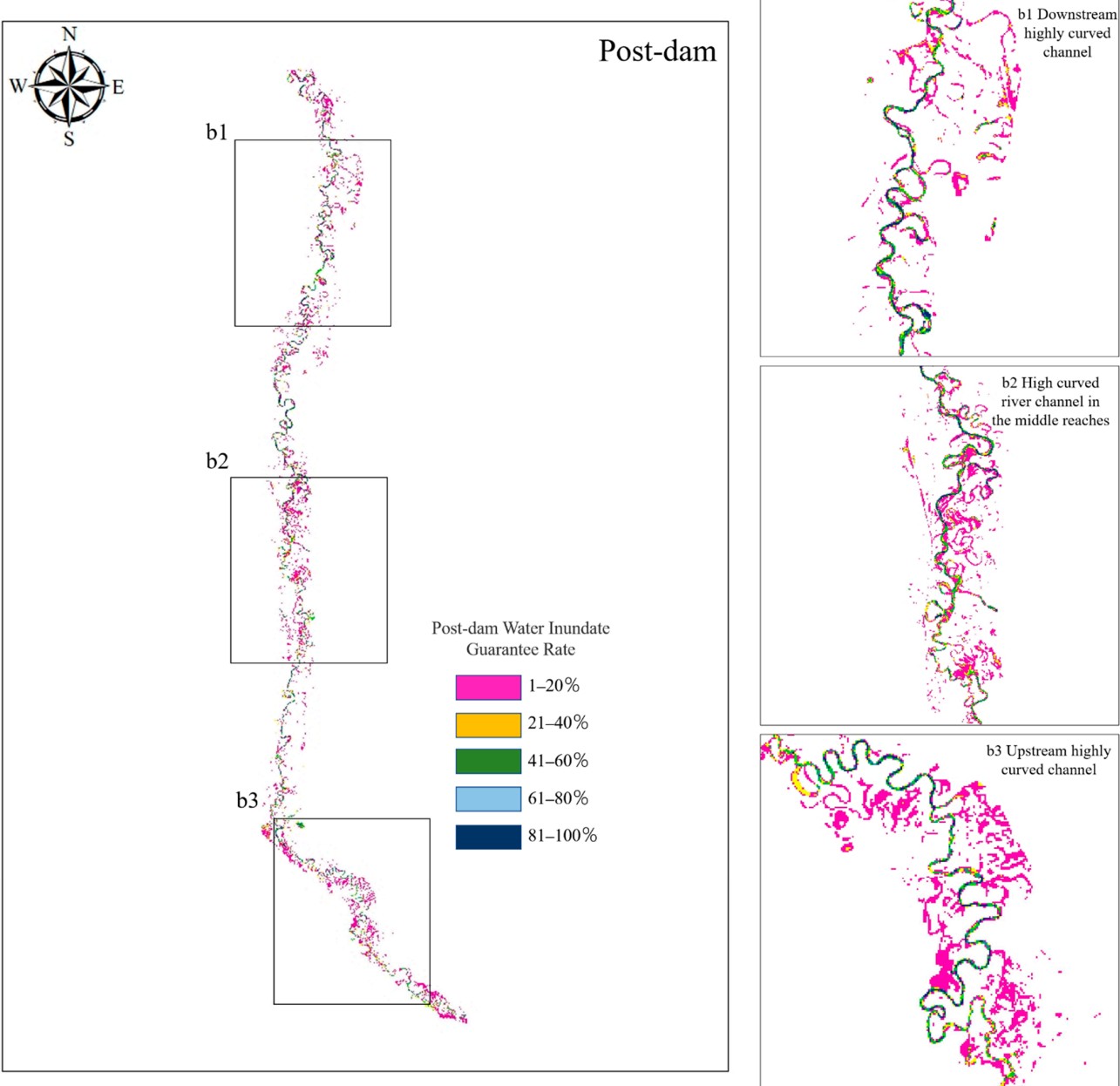

**Figure 10.** Post-dam Area distribution and typical change area under different inundation guarantee rates.

*5.4. Horizontal Distribution Characteristics of Rivers in the Affected Wetlands*

Maintaining the ecological flow in rivers and bank areas where water inundation occurs is crucial. It is not only beneficial for the stability of the ecological environment in these regions but can also increase the biodiversity in the affected regions [8]. Because of the regulation effect of the reservoir, the influences of rainfall and runoff size in the summer on the water-inundation area and WIF decrease, and the influences of extreme high/low flows are eliminated. This decreases the probability of river flooding. In other words, reservoir construction decreases the degree and frequency of water inundation in wetlands at the lower reaches after the dam construction, and the reservoir can decrease the flooding area directly by changing the flow state [40].

The variations in WIGR values in the study area are shown in Figure 11. That is, the changes in WIGR before and after the construction of the water conservancy project (see the table notes for the explanation); before the construction of the reservoir, the water

guarantee rate of different frequencies has disappeared by 16.11 km$^2$ compared with that after the construction, and it is mainly distributed in the flooding areas out of the main channel. This WIGR region disappeared after dam construction, indicating that there was no flood supply and the risk of ecosystem degradation in the wetlands was relatively high. Near-shore areas do not only decrease in WIGR; some portions exhibit an increased value and also form new WIGR regions. This indicates that the WIGR value of near-shore areas increased, and water-inundation probability increased after dam construction. This near-shore area forms part of the development region of the wetland ecosystem. Regions with basically stable WIGR values are mainly distributed at the two sides of the channel, covering 9.04 km$^2$. The WIGR in these regions changed slightly after dam construction, indicating that these regions were slightly influenced by flood supply and that the wetland ecosystem was relatively stable.

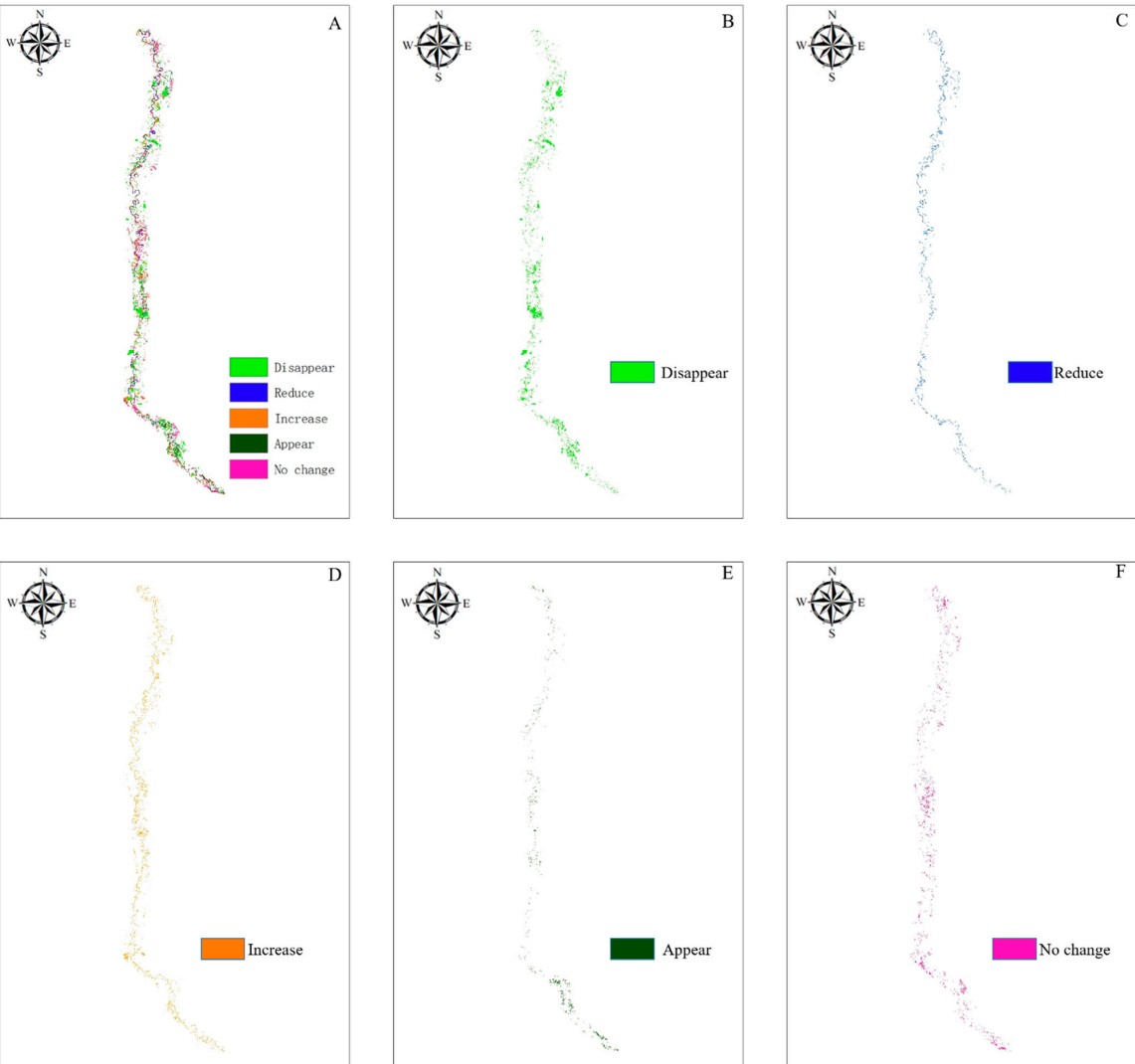

**Figure 11.** Transformation of the WIGR areas before and after dam construction. (This figure shows the change in WIGR before and after the construction of the water conservancy project, namely: (**A**): the change in the overall WIGR before and after the construction of the water conservancy project; (**B**): Indicates the area where WIGR values disappear before and after construction; (**C**): Indicates the area where the WIGR value decreases before and after construction; (**D**): Indicates the area where WIGR value increases before and after construction; (**E**): Indicates the area where WIGR values appear before and after construction; (**F**): It refers to the area where the WIGR value has no change before and after construction).

According to this transformation, reservoir regulation led to a decrease in floodplain wetlands [40]. This is mainly because the flow loss and horizontal water exchange decreased after dam construction, which had detrimental impacts on the hydrological regime in floodplain wetlands [40]. In contrast, areas with extremely high WIGR in the river increased. Since regions with high WIF were in the main channel, the effects of dam construction on the hydrological regimen in the river remained changed. The basic flow or low flow—which was regulated by the reservoir—increased the WIF, thus increasing the area with high WIGR.

The WIGR model has a broad application prospect in floodplain wetlands and flood inundated areas (coastal areas, mangrove wetlands, etc.), such as the impact of the construction of water conservancy projects on downstream wetlands, the potential contribution of flood inundation to vegetation succession, heavy metal deposition in the soil on both banks of the river, etc., which can provide new ideas and references for the study of ecosystem stability in the area where the flood meets the land. At the same time, according to the flood change in the downstream floodplain wetland and the impact of the flood on the vegetation, the number and duration of flood discharges of the upstream hydro junction can also be fed back, which will be the next research focus of this study.

## 6. Conclusions

Based on the 52 Landsat data monitoring and assessment, this study evaluated the inundation status of the Yimin River and wetlands on both banks before and after the construction of the Honghuaerji Reservoir. Combined with the analysis and discussion of the flow observation data, the following conclusions were drawn:

(1)  This study assesses the spatial impact of water conservancy project construction on floodplain wetlands based on the WIGR model. This study is the application of the WIGR model, which will have broad application prospects in floodplain wetland biodiversity, ecosystem stability, conservation area delimitation, etc., but there are also some defects such as the impact of remote sensing data cloud and rain, the number of remote sensing data, and the accuracy of hydrological data that will have an impact on the application of WIGR model in floodplain wetland;

(2)  After the construction of Honghuaerji Reservoir, the hydrological situation of the Yimin River was affected, and the runoff of the Yimin River will decrease, the predictability of the runoff will become worse, and the flow of the river will decrease, affecting the floodplain wetlands, reducing the frequency of river inundation, and resulting in a reduction in the coverage area of the inundation guarantee rate after the construction of the reservoir; the construction of the reservoir has changed the natural runoff process of the river, making the hydrological situation of the river tend to be stable and uniform, making the flow stable, resulting in an increase in the area with a very high inundation guarantee rate in the river channel;

(3)  The surrounding area of the high-curvature reaches is the main development area of floodplain wetlands, and is also the main area affected by hydrological changes;

(4)  The construction of the reservoir reduces the flow loss and the exchange of lateral water bodies, resulting in a negative impact on the hydrological status of the floodplain wetland, the area of the floodplain outside the river channel is reduced, the hydrological situation in the river channel has changed due to the influence of the dam, and the reservoir has changed. A regulated base flow or low flow will increase its inundation frequency, resulting in increased coverage of high inundation guaranteed rates.

**Author Contributions:** All authors contributed to the study conception and design. Material preparation, data collection and analysis were performed by X.D., and C.H. The first draft of the manuscript was written by C.H. and all authors commented on previous versions of the manuscript. All authors have read and agreed to the published version of the manuscript.

**Funding:** This work was funded by the National Natural Science Foundation of China, the funding number is U2102209.

**Institutional Review Board Statement:** Not applicable. This study not involving humans or animals.

**Informed Consent Statement:** Not applicable.

**Data Availability Statement:** The data set or custom code used or analyzed in the current research can be obtained from the corresponding author upon reasonable request.

**Conflicts of Interest:** The authors have no relevant financial or non-financial interests to disclose. The authors have no conflicts of interest to declare that are relevant to the content of this article. All authors certify that they have no affiliations with or involvement in any organization or entity with any financial interest or non-financial interest in the subject matter or materials discussed in this manuscript. The authors have no financial or proprietary interests in any material discussed in this article.

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
