# Peer review of "Research on the Impact of Water Conservancy Projects on Downstream Floodplain Wetlands—Taking Yimin River as an Example"

_water, doi:10.3390/w14244064_

Round 1

Reviewer 1 Report

Minor revision is required

1- Highlight your novelty and research gap

2- results are not sufficient

3- literature review should be improved

Reviewer 2 Report

General comment:

The language and organization of the manuscript are not clear. The interpretations do not necessarily follow from the data reported, and the aims and objectives of the study are not clear. It is my regret that I have deep reservations about the current version of the proposed research study. In my opinion, there are weaknesses in the research aims, methods, data processing, interpretation, and presentation of findings.

Specific remarks:

The manuscript discusses reservoir construction. Although I find the draft disappointing, I feel compelled to suggest a major revision.

1. It would be very helpful if the authors clearly articulated the aims of their study. At the end of the introduction, objectives are defined. However, not all of these issues are to the point. It would be beneficial to define the research questions more clearly and use them as the backbone of the manuscript. The reader expects to get some answers to these questions. In this manuscript, however, the answers are not presented. It seems as if the authors forgot their questions while writing the manuscript.

2. Furthermore the methods used are not explained very clearly. For example, the link between data source and processing is not evident. Specifically, three high-curvature river sections were chosen in the region to explore the influence of curvature on the wetland inundation state.

3. Figures and the table need to be explained more clearly. For example, differences in the submerged state between the highly curved river segment and the straight river segment need to be explained.

4. Have any analyses of the trends in rainfall and runoff before and after the construction of the water conservancy project been made? Is it possible to get any information on river development and compare the water properties to the properties of neighboring areas?

5. References to the journal format are not specifically provided, and a larger edition is needed.

Constructive feedback:

The results are not trustworthy in my opinion. Results and Discussion do not include any support. In fact, many processes were not measured by the authors. Similarly, general statements in the Conclusion are not supported. There is a need for further explanation for the study area section (scale is missing in Figure 6) because it seems that only a few samples were collected and there is no true replication. Please verify the method used to collect the samples. I suggest repeating the classification with a consistent criterion.

Summary:

In summary, I suggest developing a clear concept, defining objectives, explaining methods, presenting results, and answering the questions defined. I guess a complete re-writing of the manuscript is necessary.

Reviewer 3 Report

There are some comments for authors to improve the quality of manuscript as follows:

(1) In the ending of the “Introduction” section, the aim of paper should be enhanced. Meanwhile, the structure of this paper is deficit. Please add some expression in the ending of the “Introduction” section.

(2) The literature review is not comprehensive and detailed. Please add some more detailed previous studies to support your developed model. There are many good studies in the related fields, but which have their shortcomings. Please enhance your novelty of developed method. Meanwhile, please update the reference list including the most recent and relevant references. Some useful references are supplied as follows: (a) A simulation-based water-environment management model for regional sustainability in compound wetland ecosystem under multiple uncertainties. (b) A mix inexact-quadratic fuzzy water resources management model of floodplain (IQT-WMMF) for regional sustainable development of Dahuangbaowa, China. (c) A developed fuzzy-stochastic optimization for coordinating human activity and eco-environmental protection in wetland ecosystem under uncertainties. (d) A risk-simulation based optimization model for wetland reallocation on Yongding floodplain, China (e) A land-indicator-based optimization model with trading mechanism in wetland ecosystem under uncertainties.

(3) Why did the authors develop this model? What practical problems can this model deal with? and how did the authors choose various variables? All above issues should be added?

(4) Please add a general framework to rationalize the idea of this paper. Meanwhile, please add the whole index system framework before your asessment, and how to construct your assessment system.

(5) The quality of flow chat is not clear enough; meanwhile, the quality of map in Figures 3, 6, 7 and 8 are not good enough, which should be revised in a clear manner.

(6) What did the results in Figures 6, 7 and 8 display for readers? The low quality map cannot support expression in the text; meanwhile, the reason of results should be enhanced.

(7) The title is Research on the impact of water conservancy projects on downstream floodplain wetlands——Taking Yimin River as an example, however, the impact mechanism results and corresponding causes should be analyzed. Meanwhile, what is the response to the impact of water conservancy projects on downstream floodplain wetlands? All above issue should be enhanced in Result section.

(8) How did this study carry out in other application? Is there any limitation and further works, which should be added in Discussion section.

(9) The language of the paper needs improvement. For example: there are very long sentences in the manuscript that need to be revised. Meanwhile, some of the sentences also have structural issue. Moreover, numbers of grammatical errors exist in the manuscript that needs to be corrected.

(10) The Conclusion section should be revised to highlight the novelty of this paper. It is expected to include not only general description of the proposed method but also a brief summary of disadvantages of this method and some future study works. It would help the readers better understand the limitation and improvement of the proposed method. 

Round 2

Reviewer 2 Report

I thank you for helping me understand the text of the document.

Reviewer 3 Report

This version is OK.